# Development and prospective external validation of a tool to predict poor recovery at 9 months after acute ankle sprain in UK emergency departments: the SPRAINED prognostic model

Michael M Schlussel,[1] David J Keene,[2] Gary S Collins,[1] Jennifer Bostock,[3] Christopher Byrne,[4] Steve Goodacre,[5] Stephen Gwilym,[6] Daryl A Hagan,[2] Kirstie Haywood,[7] Jacqueline Thompson,[2] Mark A Williams,[8] Sarah E Lamb,[1,2] for the SPRAINED study team

For numbered affiliations see end of article.

**Correspondence to**
Dr Michael M Schlussel;
michael.schlussel@csm.ox.ac.uk

## ABSTRACT

**Objectives** To develop and externally validate a prognostic model for poor recovery after ankle sprain.

**Setting and participants** Model development used secondary data analysis of 584 participants from a UK multicentre randomised clinical trial. External validation used data from 682 participants recruited in 10 UK emergency departments for a prospective observational cohort.

**Outcome and analysis** Poor recovery was defined as presence of pain, functional difficulty or lack of confidence in the ankle at 9 months after injury. Twenty-three baseline candidate predictors were included together in a multivariable logistic regression model to identify the best predictors of poor recovery. Relationships between continuous variables and the outcome were modelled using fractional polynomials. Regression parameters were combined over 50 imputed data sets using Rubin's rule. To minimise overfitting, regression coefficients were multiplied by a heuristic shrinkage factor and the intercept re-estimated. Incremental value of candidate predictors assessed at 4 weeks after injury was explored using decision curve analysis and the baseline model updated. The final models included predictors selected based on the Akaike information criterion (p<0.157). Model performance was assessed by calibration and discrimination.

**Results** Outcome rate was lower in the development (6.7%) than in the external validation data set (19.9%). Mean age (29.9 and 33.6 years), body mass index (BMI; 26.3 and 27.1 kg/m$^2$), pain when resting (37.8 and 38.5 points) or bearing weight on the ankle (75.4 and 71.3 points) were similar in both data sets. Age, BMI, pain when resting, pain bearing weight, ability to bear weight, days from injury until assessment and injury recurrence were the selected predictors. The baseline model had fair discriminatory ability (C-statistic 0.72; 95% CI 0.66 to 0.79) but poor calibration. The updated model presented better discrimination (C-statistic 0.78; 95% CI 0.72 to 0.84), but equivalent calibration.

**Conclusions** The models include predictors easy to assess clinically and show benefit when compared with not using any model.

---

### Strengths and limitations of this study

► This is the first study to develop and externally validate a tool to predict poor recovery after ankle sprain, including a wide range of clinically relevant candidate predictors.
► Despite containing information on the outcomes of interest and numerous prognostic variables, the development data set was not originally acquired to build a prognostic model.
► The number of events in the development data set was relatively small for the number of candidate predictors examined.
► Yet, the prognostic models were developed using robust statistical methods, adjusted for overfitting and reported according to the most recent relevant guidelines available.
► Generalisability of findings is enhanced by the multicentre characteristic of the data sets used for the development and external validation of the models.

---

**Trial registration number** ISRCTN12726986; Results.

## INTRODUCTION

Ankle sprains are one of the most common musculoskeletal injuries, representing up to 5% of all emergency department (ED) attendances in the UK.[1] Despite heterogeneity in sampling frame (eg, restricted to elite athletes or excluding older people), inception and follow-up time points, studies have indicated that approximately 30% of people have persistent problems 1 year after ankle sprain.[2 3] In a large multicentre randomised clinical trial conducted in the UK, a similar proportion (30%) of participants had poor outcome at 9 months.[4] Other studies indicate

a recovery plateau at around 9 months, and residual disability after this point to be persistent.[5]

In the acute phase after a sprain, physical examination of the ankle is often difficult due to swelling and pain. Predicting prognosis at this stage is uncertain and based on clinical judgement. When concerned about the injury severity, clinicians operate a system of review within 1 week in a trauma clinic (or equivalent service), which allows some resolution of swelling and reassurance about the presence of other significant mechanical derangement.[6] The Ottawa ankle rule is also an alternative to reduce the requirement for imaging without missing important fractures.[7]

In 2008, van Rijn *et al* conducted a systematic review on the clinical pathway and prognostic factors of ankle sprain recovery and found a single eligible study concluding that high levels of sports activity have prognostic value for residual symptoms.[2] In a more recent systematic review, we have identified nine studies reporting results for baseline prognostic factors of recovery after an acute ankle sprain.[8] Age, gender, swelling, range of motion, weight-bearing ability, pain, injury severity, palpation/stress score, injury mechanism, self-reported recovery, resprain, MRI determined number of sprained ligaments and bone bruise were reported as independent predictors of poor recovery. However, almost all studies performed poorly on the risk of bias assessment, mainly due to incomplete or inadequate reporting standards for study participants, attrition, methods of assessment for predictors, confounding and statistical methods used, so results should be interpreted with caution.

To the best of our knowledge, there are no externally validated prognostic models for recovery after acute ankle sprain. Polzer *et al* developed an algorithm to help clinicians with the diagnosis and treatment of acute ankle injuries, but this is considerably based on expert judgements and does not use currently recommended methods for the development of prognostic models.[3] A robustly developed and validated prognostic model could help target treatment better and improve outcomes for people who have an ankle sprain.[9] Therefore, the development of a new prognostic model, considering a range of plausible candidate predictors, and ideally with the evaluation of its performance on an external data set (external validation), is indicated.

The aim of our study was to develop and externally validate the Synthesising a Clinical Prognostic Rule for Ankle Injuries in the Emergency Department (SPRAINED) prognostic model, to identify people at risk of poor recovery at 9 months after acute ankle sprain.

## METHODS
### Study populations and data collection
Data from the Collaborative Ankle Support Trial (CAST) were used to develop the prognostic model.[10] CAST was a pragmatic multicentre randomised controlled trial on the effectiveness of different mechanical ankle supports compared with a double-layer tubular compression bandage for managing severe ankle sprains. The trial sample comprised 584 participants aged 16 years or older, with an ankle sprain of grade 2 or 3, attending eight EDs in the UK between April 2003 and July 2005, within 7 days after their injury, and not able to fully bear weight on the injured ankle at baseline. Further data were collected at 4 and 12 weeks, and 9 months after randomisation. The CAST methods and a Consolidated Standards of Reporting Trials flow diagram are available elsewhere.[10]

To assess the model's performance in an external population, the SPRAINED prospective observational cohort was recruited. Participants were aged 16 years or above, with acute ankle sprains of any grade, attending 10 National Health Service EDs across England, within 7 days after their injury. Patients were excluded if they presented with an ankle fracture (except flake fractures <2 mm) or any other recent (<3 months) lower limb fracture. Participants were not randomised, nor did they receive any interventions other than usual care at each site. The study recruited 682 participants between July 2015 and March 2016. Data collection covered clinical and sociodemographic information assessed at ED presentation (baseline), with follow-up assessments at 4 weeks, 4 and 9 months after the initial injury, either by self-reported paper-based forms sent back to the study office by postal mail, electronic questionnaires or telephone interviews. The SPRAINED questionnaires included all variables selected as predictors in the model and the components of the outcome of interest. All participants of both studies have provided written informed consent before any data collection took place.

### Definition of outcome
A prognostic model was developed to predict 'poor recovery' at 9 months after an acute ankle sprain. Poor recovery was defined as the presence of pain, lack of confidence in the ankle (persistent feeling of giving way) or functional difficulty.[11 12] The presence of these symptoms was assessed by patient-reported responses given to specific items (P1, Q3 and Q4) of the Foot and Ankle Outcome Score (FAOS).[13] Participants who answered one or more of these questions with any of the two most extreme response options ('daily' or 'always' for P1; 'severely' or 'extremely' for Q3 or Q4) were considered to have poor outcome.

### Baseline candidate predictors
Thirty-two baseline variables were considered plausible candidate predictors of poor outcome and preselected from a pool of 170 variables available in the CAST data set (online supplementary tables 1 and 2). This initial selection was made internally by the research team, taking into account the results from our systematic literature review[8] and the conclusions from a consensus group meeting convened for the SPRAINED study, which included clinicians, medical researchers, statisticians and Patient and Public Involvement (PPI) representatives. The 32

candidate predictors included sociodemographic information (eg, age, sex, body mass index (BMI), education, employment status); preinjury quality of life, mobility and lifestyle indicators (eg, engagement in sports activities); clinical data on injury presentation; baseline (postinjury) mobility levels, pain and weight-bearing status (online supplementary table 3).

At this stage, variables were excluded or combined before statistical modelling if they had 60% or more of missing information; displayed high collinearity (r≥0.8) with another candidate predictor; presented empty or low cell counts (n<5) when tabulated against the outcome; or were the offending variable causing perfect prediction during the multiple imputation process (online supplementary table 4 and figure 1).

## Sample size considerations

It is widely recommended that the data set used to develop a prognostic tool should contain a minimum of 5–10 outcome events per variable (EPV) included as a predictor in the model.[14–19] After the exclusion of nine baseline candidate predictors for the reasons described above, 23 variables from baseline remained as candidate predictors. However, some of these predictors were categorical variables with more than two levels, so we ended with 35 candidate parameters, meaning the EPV ratio was approximately 3.

As to the best of our knowledge this is the first study aiming to develop prediction models to assess the risk of poor recovery after an acute ankle sprain, we opted for relaxing the EPV rule in favour of including more potentially important predictors. Nevertheless, we adopted several strategies to minimise bias and overfitting, as described below.

## Descriptive analysis

Baseline and 4-week follow-up characteristics of the CAST and SPRAINED participants were summarised using means, SDs and ranges for continuous variables, or counts and percentages for categorical variables. Inspection of extreme values (outliers) took place to confirm whether they were clinically plausible and visual assessment of data distribution for continuous predictors in both data sets was conducted. No formal statistical tests were performed to compare the values between the studies.

## Prognostic model development

Using logistic regression, we developed the prognostic model to predict the probability of poor recovery. We performed multiple imputation using chained equations (MICE)[20] to handle missing data, with 50 imputed data sets created. Continuous variables were kept as continuous to avoid loss of prognostic information,[21] and the shape of their relationship with the outcome studied and modelled with non-linear functions such as fractional polynomials (FP) where appropriate.[22] As several continuous variables were included in the models, we used the multivariable fractional polynomial (MFP) algorithm.[23 24]

Multiple imputation and FPs were combined using the *mfpmi* function in Stata.[25] The estimated regression parameters (coefficients and variances) were combined over the 50 imputed data sets using Rubin's rule.[26 27] After identifying the best transformation terms for continuous variables, the final model included predictors (and respective transformations, where applicable) selected from the full multivariable model with all candidate predictors based on the Akaike information criterion (AIC; equivalent to p<0.157).[28] To adjust for overfitting, due to small EPV, we multiplied all regression coefficients by the heuristic shrinkage factor,[29] then re-estimated the intercept. All model assumptions were checked and differences between incomplete and imputed data sets inspected. Imputed data from all 584 participants were included in all analyses.

## Incremental value analysis and model update

In addition to the baseline predictors, 14 additional variables from the CAST 4 weeks' follow-up questionnaire were also selected as potential predictors that could increase the model's prognostic ability (online supplementary table 3). First, all additional 4 weeks' candidate predictors were included together in the final baseline model and only those achieving p<0.157 were considered for inclusion in the updated model (ie, a model including baseline and 4 weeks' predictors). Finally, the updated model was compared with the original baseline model using decision curve analysis (DCA) plots to determine whether the inclusion of additional predictors reflected in increased net benefit.[30 31]

## External validation (model performance evaluation)

We assessed the model performance in the prospectively collected SPRAINED cohort. Missing data in the SPRAINED cohort were handled using MICE, creating 50 imputed data sets. Performance was evaluated by assessing calibration and discrimination.

Calibration is the agreement between observed and predicted probabilities of poor outcome. Calibration was assessed graphically using calibration plots, with observed risks plotted on the y-axis against predicted risks on the x-axis.[32 33] The calibration plot was created by regressing the outcome on the predicted probability using a locally weighted scatter plot smoother (lowess). The calibration plot was also supplemented with estimates of the calibration slope and intercept. Models with perfect calibration will have a calibration slope of 1 and intercept 0 (ie, prediction lying on the 45° line). Calibration plots followed the recommendations of overlaying calibration curves from each imputed data set.[34]

Discrimination reflects the ability of the model to distinguish between participants who did and did not experience an event during the study period. Discrimination was assessed using the C-statistic, where a value of 0.5 represents chance and one represents perfect discrimination.[35] Finally, to estimate the benefit of using the developed models, the patients were ranked according

to their estimated risks. These were used to calculate the number of people per 1000 identified as being at high risk according to selected thresholds and how many of these went on to present the outcomes compared with not using the model. Individual probabilities of developing the outcomes were estimated by applying the developed prognostic models to each participant in the SPRAINED imputed data sets. We assessed the performance of both the baseline and updated models using imputed data from all 682 participants.

## Patient involvement

A PPI representative was involved in the study from the beginning, providing advice on key aspects of the study design, including the definition of the research question, choice of the outcome and selection of relevant candidate predictors during the consensus group meeting.

They will be consulted for the public dissemination of any product arriving from this research.

## Reporting

We followed the Transparent Reporting of a Multivariable Prediction Model for Individual Prognosis or Diagnosis (TRIPOD) statement for the reporting of our study.[36]

## RESULTS

Baseline characteristics for the CAST (development) and SPRAINED (validation) cohorts are summarised in table 1. On average, participants were slightly older in SPRAINED than in CAST. Participants in SPRAINED had an average BMI within the overweight category, likewise those in CAST. The mean pain scores when resting or bearing weight on the ankle of SPRAINED participants

**Table 1**  Baseline characteristics of the participants in the CAST trial and SPRAINED prospective observational cohort

| Variable | CAST trial | | SPRAINED cohort | |
|---|---|---|---|---|
| | Mean (SD) | Min–Max | Mean (SD) | Min–Max |
| Age (years) | 29.88 (10.77) | 16–72 | 33.62 (13.38) | 16–89 |
| Height (m) | 1.73 (0.98) | 1.47–2.01 | 1.72 (1.02) | 1.50–2.01 |
| Weight (kg) | 78.56 (15.44) | 39.92–133.36 | 80.44 (18.13) | 44.50–180 |
| Body mass index (kg/m$^2$) | 26.34 (5.19) | 16.07–53.77 | 27.08 (5.70) | 17.31–64.30 |
| Pain when resting (score) | 37.75 (23.49) | 0–100 | 38.50 (22.50) | 0–100 |
| Pain when bearing weight (score) | 75.42 (19.61) | 0–100 | 71.30 (21.00) | 0–100 |
| | Frequency | % | Frequency | % |
| Sex | | | | |
| Male | 337 | 57.71 | 327 | 47.95 |
| Female | 247 | 42.29 | 355 | 52.05 |
| Days from injury to assessment | | | | |
| 0–2 | 118 | 44.87 | 614 | 90.03 |
| 3 or more | 145 | 55.13 | 68 | 9.97 |
| Able to bear weight at baseline assessment | | | | |
| No | 446 | 77.03 | 179 | 26.44 |
| Yes | 133 | 22.97 | 498 | 73.56 |
| Recurrent sprain | | | | |
| No | 517 | 90.38 | 583 | 91.38 |
| Yes | 55 | 9.62 | 55 | 8.62 |
| Current employment | | | | |
| None | 132 | 22.60 | 161 | 23.68 |
| Part time | 92 | 15.75 | 92 | 13.53 |
| Full time | 360 | 61.64 | 427 | 62.79 |
| Injury mechanism | | | | |
| At home | 99 | 18.00 | 144 | 21.56 |
| Practising sports | 203 | 36.91 | 230 | 34.43 |
| At work | 79 | 14.36 | 91 | 13.62 |
| Outside, in public | 169 | 30.73 | 203 | 30.39 |

CAST, Collaborative Ankle Support Trial; SPRAINED, Synthesising a Clinical Prognostic Rule for Ankle Injuries in the Emergency Department.

**Table 2** Outcome and respective symptoms component rates and proportion of missing data in the CAST trial and SPRAINED prospective observational cohort

|  | Pain (%) | Lack of confidence (%) | Instability (%) | Poor recovery (%) | Missing data (%) | Total |
|---|---|---|---|---|---|---|
| CAST | 84 (14.4) | 42 (7.2) | 67 (11.5) | 116 (19.9) | 144 (24.7) | 584 |
| SPRAINED | 3 (0.4) | 23 (3.4) | 37 (5.4) | 46 (6.7) | 155 (22.7) | 682 |

Poor recovery defined as the presence of one or more of the following symptoms: pain, lack of confidence or instability/difficulty with the ankle.
CAST, Collaborative Ankle Support Trial; SPRAINED, Synthesising a Clinical Prognostic Rule for Ankle Injuries in the Emergency Department.

were also similar to those observed for CAST participants. Differently from CAST, in SPRAINED about half of participants were female, the majority presented to an ED within 2 days from injury for assessment and were able to bear some weight on their injured ankles (table 1).

Table 2 shows the rates of poor recovery in the CAST trial and SPRAINED cohort data sets, as well as the number of its component symptoms, at 9 months after injury. There was a lower rate of poor recovery in the SPRAINED cohort than observed in the CAST trial, but the percentage of missing data for the outcome was similar in both studies.

Table 3 displays the summary of the final multivariable models (predictor's coefficients, respective 95% CIs and p values). Seven of the 23 baseline candidate predictors were selected for inclusion in the baseline model: age, BMI, pain when resting, pain when bearing weight, days from injury to assessment, ability to bear weight and whether or not the injury was a recurrent sprain. The best fit for all continuous predictors was linear transformations (mean subtractions) and was later incorporated

into the model by updating the intercept accordingly (online supplementary table 5).

Linear terms selected by the MFP for continuous predictors were: age -29.88; BMI -26.32; pain when resting -37.75; pain when bearing weight -75.40; pain when bearing weight at 4 weeks after injury -36.23.

Only pain when bearing weight on the injured ankle at 4 weeks after injury was included in the updated model (baseline plus 4-week predictor) (table 3). By inspecting the DCA plot shown in figure 1 it is possible to see a clear net benefit gain over the entire range of thresholds when using the updated prognostic model in comparison to the baseline model or considering all patients (or no patient) at risk of having poor recovery after an acute ankle sprain.

Shrinkage suggested both prognostic models (baseline and updated) had predictor-outcome associations that were too large. The heuristic shrinkage factor for the coefficients of the predictors in the baseline prognostic model was 0.71. For the updated model (baseline plus 4 weeks' predictor), the estimated heuristic shrinkage

**Table 3** Summary of the final baseline and updated (baseline plus 4 weeks' predictor) logistic regression models and respective shrunk coefficients and intercepts

| Predictors | Baseline model | | | | Updated model (baseline plus 4 weeks' predictors) | | | |
|---|---|---|---|---|---|---|---|---|
|  | Coefficient | 95% CI | P values | Shrunk coefficient | Coefficient | 95% CI | P values | Shrunk coefficient |
| Age | 0.027 | 0.006 to 0.048 | 0.014 | 0.019 | 0.018 | −0.005 to 0.040 | 0.127 | 0.015 |
| BMI | 0.031 | −0.014 to 0.076 | 0.178 | 0.022 | 0.025 | −0.022 to 0.072 | 0.292 | 0.021 |
| Pain when resting | 0.016 | 0.005 to 0.027 | 0.005 | 0.011 | 0.010 | −0.002 to 0.022 | 0.107 | 0.008 |
| Pain when bearing weight | 0.019 | 0.004 to 0.035 | 0.016 | 0.014 | 0.014 | −0.002 to 0.030 | 0.092 | 0.012 |
| Pain when bearing weight 4 weeks after injury | – | – | – | – | 0.022 | 0.012 to 0.032 | <0.001 | 0.018 |
| Days from injury to assessment (reference: 0–2) | | | | | | | | |
| 3 or more | 0.854 | 0.068 to 1.640 | 0.034 | 0.605 | 0.702 | −0.117 to 1.520 | 0.092 | 0.591 |
| Able to bear weight at baseline (reference: No) | | | | | | | | |
| Yes | −0.792 | −1.376 to −0.207 | 0.008 | −0.561 | −0.802 | −1.412 to −0.192 | 0.010 | −0.676 |
| Recurrent sprain (reference: No) | | | | | | | | |
| Yes | 1.180 | 0.417 to 1.944 | 0.003 | 0.836 | 1.170 | 0.386 to 1.953 | 0.004 | 0.985 |
| Intercept | −1.580 | −2.152 to −1.008 | <0.001 | −1.363 | −1.543 | −2.128 to −0.958 | <0.001 | −1.420 |

BMI, body mass index.

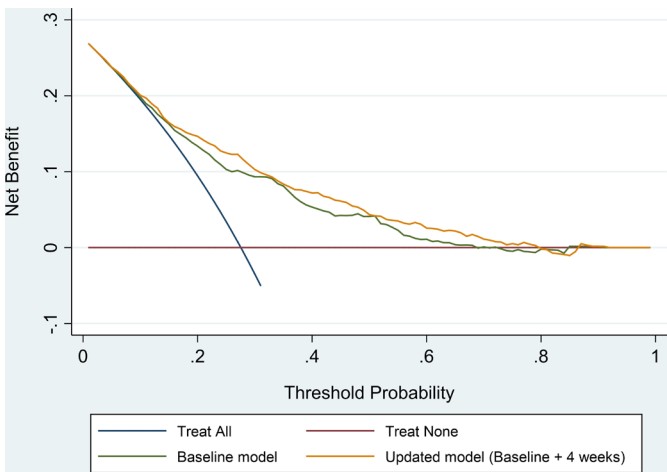

**Figure 1** Decision curve analysis for the baseline and updated (baseline plus 4 weeks' predictor) prognostic models.

factor was 0.84. The shrunk coefficients and intercepts for the final models are presented in table 3.

Overall, discrimination of the baseline model was fair, with a C-statistic of 0.72 (95% CI 0.66 to 0.79). Calibration of the baseline prognostic model in the external validation data set was poor though, as can be evidenced by inspecting the calibration plot with overlaid calibration lines from the 50 imputed data sets (figure 2). The calibration slope was 1.13 (95% CI 0.76 to 1.5) and the calibration intercept was −0.71 (95% CI −0.98 to −0.44). The updated model (baseline plus 4 weeks' predictor) presented better discriminatory ability in the SPRAINED data set than the baseline model (C-statistic=0.78; 95% CI 0.72 to 0.84), but equivalent calibration, with an intercept closer to 0 (−0.51; 95% CI −0.78 to −0.24) and slope slightly further from 1 (1.17; 95% CI 0.86 to 1.48).

Table 4 shows how many of 1000 people would be identified as being at high risk (based on thresholds of 5%, 10%, 15% and 20%) using the developed prognostic models, and how many of these would actually present poor recovery 9 months after an acute ankle sprain. There seems to be little difference between the baseline and updated models, with both identifying similar numbers of patients who would experience a poor outcome after an acute ankle sprain. However, less patients are deemed at high risk by using the updated model for (less false positives) across all thresholds of predicted probability, suggesting that reassessing the patients at 4 weeks after the injury might be beneficial to a more accurate prediction of their probability of poor outcome. Using any of the models is clearly beneficial when compared with not using any model (ie, considering all patients—or no patients—as high risk of developing poor outcome).

## DISCUSSION

We developed a prognostic model to predict a composite outcome representing the presence of at least one of the following symptoms at 9 months after an acute ankle sprain: pain, functional difficulty or lack of confidence in the ankle. The model presented fair discriminatory ability in a prospective cohort composed for the models external validation, but poor calibration. Including an additional variable collected at 4 weeks after the injury (pain when bearing weight on the injured ankle) improved the discriminatory ability of the model. The models include predictors that are easy to assess and provide reasonable predictions of poor recovery for patients with acute ankle sprain.

In a recent systematic review, we have reported that some of the variables selected for inclusion in our

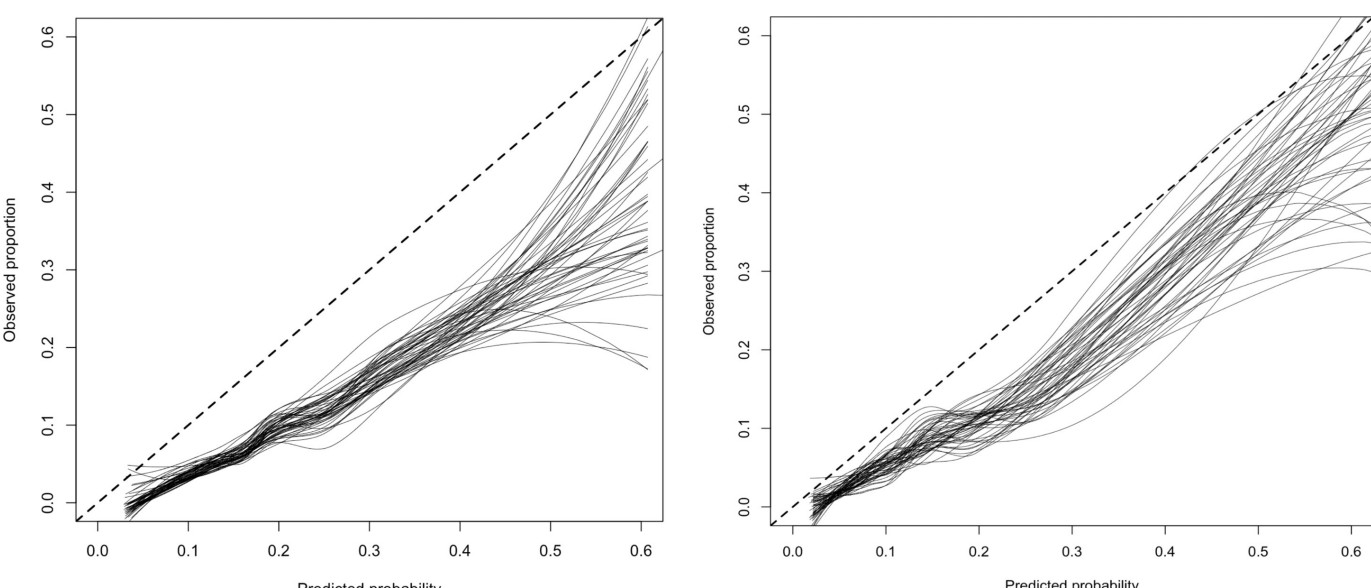

**Figure 2** Calibration plots for the baseline (left) and updated (right) SPRAINED prognostic models, overlaying calibration lines derived from the analyses of 50 imputed data sets. SPRAINED, Synthesising a Clinical Prognostic Rule for Ankle Injuries in the Emergency Department.

**Table 4** Model performance (numbers at risk and outcomes identified) at varying risk thresholds for 1000 patients

| Selected thresholds | Number of patients at risk | | Number of events | |
| --- | --- | --- | --- | --- |
| | High risk | Low risk | Identified | Not identified |
| Consider all high risk | 1000 | 0 | 85 | 0 |
| Predicted probability as per baseline model | | | | |
| ≥5% | 971 | 39 | 85 | 0 |
| ≥10% | 797 | 203 | 74 | 11 |
| ≥15% | 543 | 457 | 63 | 22 |
| ≥20% | 351 | 649 | 52 | 33 |
| Predicted probability as per updated model (baseline plus 4 weeks' predictor) | | | | |
| ≥5% | 882 | 118 | 85 | 0 |
| ≥10% | 517 | 483 | 71 | 14 |
| ≥15% | 358 | 642 | 56 | 29 |
| ≥20% | 259 | 741 | 41 | 44 |

prognostic models have been previously identified as important predictors of short, medium or long-term recovery after ankle sprain.[8] According to O'Connor et al, age and weight-bearing ability are predictors of ankle function, as measured by the Karlsson function score, both at 4 weeks and 4 months after injury.[37] Akacha et al also demonstrated that age was an important predictor of slower and incomplete recovery after ankle sprain, as measured by the FAOS.[38] The magnitude of pain at rest at 3 months has also been shown to have prognostic value for poorer self-reported recovery at 12 months after ankle sprain by van Middelkoop et al.[39] On the other hand, findings regarding recurrence of ankle sprain are conflicting. Medina McKeon et al reported that recurrent ankle sprain was not a significant predictor of time to return to play after an ankle injury.[40] This is contrary to reports of an association between recurrent sprains and chronic ankle instability reported in a systematic review conducted by Pourkazemi et al.[41] One possible explanation for these contradictory results may be the nature of the outcomes investigated in each study. When more subjective aspects of recovery (such as ankle function or instability) are considered in the definition of the endpoint, like in the present study, respraining the ankle seems to be an important predictor of recovery.

The inclusion of BMI in the prognostic model is another issue that deserves consideration. Although not statistically significant in the final multivariable logistic regression analysis, according to AIC (p<0.157), we have decided to keep BMI in the model for several reasons. First, this decision prevented another round of predictor selection, which could increase overfitting. The model building process was not solely based on statistical rationale, and BMI was considered to be an important predictor by clinicians during our consensus group meeting. BMI is an easy to assess surrogate measure of body weight that is frequently collected at clinical routine

and one that most patients know how to calculate themselves. Finally, its inclusion does not add much complexity to the models.

To the best of our knowledge, this is the first study to develop and externally validate a prognostic model to predict a clinically relevant outcome in people with acute ankle sprains, and exploring a wide range of clinically plausible candidate predictors. We used robust statistical methods to select the predictors and assess the model's performance in a large external prospective cohort. Generalisability of the findings is enhanced by the multicentre feature of both the CAST and SPRAINED samples that represented a range of district general and major trauma centres. The observational cohort we prospectively recruited for SPRAINED is representative of patients presenting to EDs in the UK. We followed the most recent and complete guidelines available on the reporting of prognostic model development,[36] and applied recommended methods to minimise overfitting. For example, continuous variables, whenever possible, were kept as continuous to avoid loss of information. Non-linear relationships were investigated using the best variables transformation found by MFPs. The study included an internal correction for model optimism (shrinkage of regression coefficients and re-estimation of intercepts) as well as a prospective external validation phase. The amount of missing data in the external validation data set, which is commonplace in studies of this nature, was considerably smaller than that observed in the development data set. Finally, we performed missing data imputation to produce a set of 50 complete data sets and enable robust analyses.

Limitations of the SPRAINED study are acknowledged. First, data used to develop the prognostic models were from a prior randomised controlled trial (CAST), so were not originally intended to fulfil this aim. However, the CAST cohort did represent the best data set available, with information on the symptoms and clinical events of

interest, and a wide range of the candidate prognostic variables considered to have predictive ability. Second, the CAST data set used to develop the prognostic model was relatively small when considering the number of candidate predictors included in the analysis.[14–19] As previously highlighted, the low EPV observed might have contributed to the optimism found for both models (baseline and updated) and, therefore, to their poor calibration on the external validation data set. Third, the amount of missing data in the development data set prevented the inclusion of a number of candidate predictors, even before the process of data imputation, to avoid instability of the imputation models. Therefore, some important predictors could have conceivably been missed in the development phase of the SPRAINED study. Finally, the rates of poor outcome in the SPRAINED cohort were lower than in the CAST trial and those reported in previous systematic reviews.[2 3] These variations in poor outcome rates and clinically important differences in baseline characteristics included in the prognostic model (such as days from injury to clinical assessment and ability to bear weight on the injured ankle) highlight the issue of different sampling frames.

Clinical examination of acute ankle sprain is challenging as tolerance of physical examination tests is often poor due to pain and swelling. Imaging is often not routinely available. A prognostic tool could enable better targeting of treatments such as immobilisation casts, which although effective can be inconvenient to patients, to those deemed at low risk of poor outcome. On the other hand, it has the potential to help clinicians targeting treatments such as surgery and physiotherapy to patients who are at highest risk of poor outcome.

The SPRAINED prognostic model benefits from including predictors that are easy to measure, and usually assessed in clinical routine. Thus, given the discussed limitations in its predictive performance, we suggest that its value would be in assisting the clinician to estimate the probability of a poor outcome, instead of being used as a decision-making tool in isolation. Improved predictive performance of the models with the addition of information on pain when bearing weight at 4 weeks indicates that reassessment of prognosis after the acute phase is worth consideration for patients initially deemed to have elevated probability of delayed recovery. Besides, as it is an easy-to-use instrument, patients themselves can estimate their probability of poor outcome and gain some reassurance in their decisions to seek for further medical assistance or not.

If implemented in clinical practice, clinicians should be aware that there is a degree of uncertainty associated to the calculated risk of poor outcome when using the SPRAINED prognostic model. This uncertainty can lead to over or under-referral of patients to review clinics or referral treatment such as physiotherapy. Future work could examine how well the model performs in comparison (or addition) to the clinician impression. Moreover, we recommend further research to evaluate the impact of using the SPRAINED prognostic model in clinical practice to predict patient outcomes and to assess the acceptability and uptake of the tool by clinicians in the EDs.

In conclusion, the SPRAINED prognostic models performed reasonably and despite some miscalibration show benefit in identifying patients at high risk of poor outcome after an acute ankle sprain. The models may assist clinical decision-making when assessing and advising people with ankle sprains in the ED setting and when deciding on ongoing management. The models benefit from using predictors that are simple to obtain during routine clinical assessment.

**Author affiliations**
[1]Centre for Statistics in Medicine, Nuffield Department of Orthopaedics, Rheumatology and Musculoskeletal Sciences, University of Oxford, Oxford, UK
[2]Centre for Rehabilitation Research, Nuffield Department of Orthopaedics, Rheumatology and Musculoskeletal Sciences, University of Oxford, Oxford, UK
[3]Patient and Public Involvement, Quality and Outcomes of Person-Centred Care Policy Research Unit, Canterbury, UK
[4]Faculty of Health and Human Sciences, University of Plymouth, Plymouth, UK
[5]School of Health and Related Research, University of Sheffield, Sheffield, UK
[6]Oxford Trauma, Nuffield Department of Orthopaedics, Rheumatology and Musculoskeletal Sciences, University of Oxford, Oxford, UK
[7]Warwick Research in Nursing, Division of Health Sciences, Warwick Medical School, University of Warwick, Coventry, UK
[8]Department of Sport, Health Sciences and Social Work, Oxford Brookes University, Oxford, UK

**Acknowledgements** We thanks Professor Richard Riley, Professor Kevin Mackway-Jones and Professor Suzanne McDonough from the Independent Study Steering Committee for their invaluable comments, support and advise throughout the study. We also acknowledge the contributions of Vicki Barber and the staff members at Oxford Clinical Trials Research Unit and Centre for Rehabilitation Research for their support in delivering the SPRAINED study; the members of the consensus group meeting, including the PPI representatives; and all those who have in some way contributed to the conception, conduction and reporting of the SPRAINED study.

**Collaborators** The SPRAINED study team members are: SEL (chief investigator); DJK (study lead); GSC, MAW, SG, Matthew Cooke, SG, Phil Hormbrey, David Wilson, JB (coinvestigators); DAH (study administrator); Damian Haywood (senior study manager); JT, CB (research physiotherapists); MMS (study statistician); Philip Hormbrey, Susan Dorrian, SG, Victoria Stacey, Tim Coats, Sarah Wilson, Jason Kendall, David Clarke, Antoanela Colda, Deborah Mayne (principal investigators and their clinical and research teams at collaborating recruitment centres); KH (consultation and senior facilitation of the consensus meeting).

**Contributors** MMS analysed and interpreted the data, and led the writing of the manuscript. DJK had substantial contribution in data acquisition, analysis and interpretation. GSC had substantial contribution in the study conception and design, data analysis and interpretation. JB, SG and KH had substantial contribution in the study conception and design. CB, DAH and JT had substantial contribution in the data acquisition. MAW had substantial contribution in the study conception and design and data acquisition. SEL was responsible for the study conception and design, and had substantial contribution in data interpretation. All authors revised and approved the final version of the manuscript.

**Funding** The SPRAINED study was funded by the National Institute for Health Research (NIHR) Health Technology Assessment programme (project number 13/19/06). Supported by the NIHR Biomedical Research Centre, Oxford, and the NIHR Fellowship programme (DJK, PDF-2016-09-056). SEL receives funding from the National Institute for Health Research (NIHR) Collaboration for Leadership in Applied Health Research and Care Oxford at Oxford Health NHS Foundation Trust.

**Disclaimer** The views and opinions expressed therein are those of the authors and do not necessarily reflect those of the Health Technology Assessment programme, NIHR, NHS or the Department of Health and Social Care.

**Competing interests** None declared.

**Patient consent** Obtained.

**Ethics approval** Ethics approval was from the National Research Ethics Committee (REC) (London–Chelsea), REC number 15/LO/0538, on 10 April 2015. The study protocol was registered on 30 April 2015 (registry number ISRCTN12726986).

**Provenance and peer review** Not commissioned; externally peer reviewed.

**Data sharing statement** All data requests should be submitted to the corresponding author for consideration. Access to anonymised data may be granted following review. Exclusive use will be retained until the publication of major outputs.

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
