## [Reviewer comments · BMJ Open]

This paper was submitted to a another journal from BMJ but declined for publication following peer review. The authors addressed the reviewers' comments and submitted the revised paper to BMJ Open. The paper was subsequently accepted for publication at BMJ Open.

(This paper received three reviews from its previous journal but only two reviewers agreed to published their review.)

ARTICLE DETAILS

TITLE (PROVISIONAL)	DEVELOPMENT AND PROSPECTIVE EXTERNAL VALIDATION OF A TOOL TO PREDICT POOR RECOVERY AT NINE MONTHS AFTER ACUTE ANKLE SPRAIN IN UK EMERGENCY DEPARTMENTS: THE SPRAINED PROGNOSTIC MODEL
AUTHORS	Schlüssel, Michael; Keene, David; Collins, Gary; Bostock, Jennifer; Byrne, Christopher; Goodacre, Steve; Gwilym, Stephen; Hagan, Daryl; Haywood, Kirstie; Thompson, Jacqueline; Williams, Mark; Lamb, Sarah

VERSION 1 – REVIEW

REVIEWER	Fereshteh Pourkazemi The University of Sydney Australia we (our research team) are investigating the impact of pain on the development of chronic ankle instability. I request the editor to dismiss my review if they believe there is a conflict of interest in the above statement.
REVIEW RETURNED	07-Apr-2018

GENERAL COMMENTS	Dear Authors, I read your article with great interest as you are addressing one of the most important questions in this field of research. Congratulations on your study. I, however, believe your paper is written with a significant focus on statistics and fails to demonstrate the clinical importance of your study. In your abstract, none of the predictors included in the model are even mentioned but your statistical model is explained in details. You mention CAST and SPRAINED in the abstract while the reader has no information on these acronyms or what your previous study was. The introduction doesn't demonstrate the gap in the literature well because multiple studies published over the recent years investigating the potential predictors of CAI are not discussed. You should lead your reader to the aim of your study by discussing the current literature more in depth.
--

	In the methods section, please ensure to add a statement demonstrating that patients/participants provided consent prior to the commencement of the study. In addition, please provide further details on your potential predictors and how they were measured (e.g.; how did you measure the quality of life? or pain? Was it a numerical scale or a visual scale?). Such information is required to allow replication of your study. If you are limited by the word count, please provide a supplementary table and provide information on the potential predictors. Please keep in mind that we want your research to be translated into clinical practice; therefore, it should be meaningful to clinicians. The result section can be summarised and you can avoid repeating the information provided in tables. Tables should be self-explanatory and if you have provided information on your findings in details, you are not required to re-state the findings in a table. Thank you
--	--

REVIEWER	Adam B. Rosen University of Nebraska at Omaha, USA
REVIEW RETURNED	16-Apr-2018

GENERAL COMMENTS	Thank you for allowing me to review this well-written manuscript. The authors attempt to create a clinical tool capable of determining who may be at risk of having poor outcomes after an initial ankle sprain. I think this study was well-done and has significant clinical implications. However, I do have some concerns and a few comments to assist in improving the manuscript for researchers and clinicians. Methods Consider adding a flow diagram to both the patients included in the model as well as the process of selection of predictors. I think these visual representations may assist the reader in following the in-text iterative processes. 156-157: I think a simple, independent samples t-test to look at differences in the data sets would be appropriate, especially due to the CAST including individuals "with an ankle sprain of grade 2 or 3" while the SPRAINED cohort included any ankle sprain. 179-181: What was the rationale for selecting a p value of 0.157? 217-225: Going back to my previous point, I think this does warrant statistical tests to determine differences across groups. 236-237 – I'm not sure if this is a valid enough of a reason to include BMI in the final model. While, BMI is a relatively simplistic way of measuring body composition, plenty of research has demonstrated its flaws. I'd like some more supportive objective commentary on its' inclusion, especially considering its relatively small contribution to the model. Discussion: I think the discussion could be developed a bit more. Some discussion on the individual predictors themselves would be appropriate.
---

VERSION 1 – AUTHOR RESPONSE

Reviewer: 1

Dear Authors,

I read your article with great interest as you are addressing one of the most important questions in this field of research. Congratulations on your study.

We thank the reviewer's kind words.

I, however, believe your paper is written with a significant focus on statistics and fails to demonstrate the clinical importance of your study.

We thank the reviewer for this comment. Indeed, we have focused the manuscript on the statistical aspects of our work. This was mainly because, as far as we know, this is the first study reporting the development of a prognostic model for poor recovery after an acute ankle sprain, so future research could benefit from a detailed description of the methods employed in the model development. However we acknowledge the point raised by the reviewer. Thus, to address the clinical application of the prognostic model we have added the following paragraphs to the discussion section:

“Clinical examination of acute ankle sprain is challenging as tolerance of physical examination tests is often poor due to pain and swelling. Imaging is often not routinely available. A prognostic tool could enable better targeting of treatments such as immobilisation casts, which although effective can be inconvenient to patients, to those deemed at low risk of poor outcome. On the other hand, it has the potential to help clinicians targeting treatments such as surgery and physiotherapy to patients who are at highest risk of poor outcome.

The SPRAINED prognostic model benefits from including predictors that are easy to measure, and usually assessed in clinical routine. Given the hereby discussed limitations in its predictive performance, we suggest that its value would be in assisting the clinician to estimate the probability of a poor outcome, instead of being used as a decision making tool in isolation. Improved predictive performance of the models with the addition of information on pain when bearing weight at 4 weeks indicates that re-assessment of prognosis after the acute phase is worth consideration for patients initially deemed to have elevated probability of delayed recovery. Besides, as it is an easy-to-use instrument, patients themselves can estimate their probability of poor outcome and gain some reassurance in their decisions to seek for further medical assistance or not.

If implemented in clinical practice, clinicians should be aware that there is a degree of uncertainty associated to the calculated risk of poor outcome when using the SPRAINED prognostic model. This uncertainty can lead to over or under referral of patients to review clinics or referral treatment such as physiotherapy. Future work could examine how well the model performs in comparison (or addition) to the clinician impression. Moreover, we recommend further research to evaluate the impact of using the SPRAINED prognostic model in clinical practice to predict patient outcomes and to assess the acceptability and uptake of the tool by clinicians in the EDs.”

In your abstract, none of the predictors included in the model are even mentioned but your statistical model is explained in details.

We thank the reviewer for this comment. We have adjusted the abstract to provide a more balanced summary of our work.

You mention CAST and SPRAINED in the abstract while the reader has no information on these acronyms or what your previous study was.

We thank the reviewer for noticing that. We have edited the abstract and removed any undefined acronyms.

The introduction doesn't demonstrate the gap in the literature well because multiple studies published over the recent years investigating the potential predictors of CAI are not discussed. You should lead your reader to the aim of your study by discussing the current literature more in depth.

We thank the reviewer for this helpful suggestion. Indeed our introduction section was brief. We have expanded the introduction to provide additional information from the published studies on the prognostic factors for recovery after ankle sprain and to contextualise the reader on the importance of conducting the present study. For that, we have included the following paragraphs:

“In 2008, Van Rijn et al conducted a systematic review on the clinical pathway and prognostic factors of ankle sprain recovery and found a single eligible study concluding that high levels of sports activity have prognostic value for residual symptoms.[2] In a more recent systematic review, we have identified nine studies reporting results for baseline prognostic factors of recovery after an acute ankle sprain.[8] Age, gender, swelling, range of motion, weight bearing ability, pain, injury severity, palpation/stress score, injury mechanism, self-reported recovery, re-sprain, MRI determined number of sprained ligaments and bone bruise were reported as independent predictors of poor recovery. However, almost all studies performed poorly on the risk of bias assessment, mainly due to incomplete or inadequate reporting standards for study participants, attrition, methods of assessment for predictors, confounding and statistical methods used, so results should be interpreted with caution.

To the best of our knowledge, there are no externally validated prognostic models for recovery after acute ankle sprain. Polzer et al. developed an algorithm to help clinicians with the diagnosis and treatment of acute ankle injuries, but this is considerably based on expert judgements and do not use currently recommended methods for the development of prognostic models.[9] A robustly developed and validated prognostic model could help to target treatment better and improve outcomes for people who have an ankle sprain.[10] Therefore, the development of a new prognostic model, considering a range of plausible candidate predictors, and ideally with the evaluation of its performance on an external dataset (external validation), is indicated.”

In the methods section, please ensure to add a statement demonstrating that patients/participants provided consent prior to the commencement of the study.

We thank the reviewer for this advice and have included the following statement in the methods section of the manuscript:

“All participants of both studies have provided written informed consent before any data collection took place.”

In addition, please provide further details on your potential predictors and how they were measured (e.g.; how did you measure the quality of life? or pain? Was it a numerical scale or a visual scale?). Such information is required to allow replication of your study. If you are limited by the word count, please provide a supplementary table and provide information on the potential predictors. Please keep in mind that we want your research to be translated into clinical practice; therefore, it should be meaningful to clinicians.

We thank the reviewer for this suggestion and agree that the inclusion of such information will enhance the completeness and transparency of our report. Therefore, we have added supplementary tables with information on all available variables in the development dataset (Supplementary Tables 1 and 2). In conjunction with the Table describing the characteristics of the pre-selected baseline and 4-

weeks candidate predictor variables (Supplementary Table 3), researchers should be able to replicate the methods used to develop the SPRAINED prognostic model.

The result section can be summarised and you can avoid repeating the information provided in tables. Tables should be self-explanatory and if you have provided information on your findings in details, you are not required to re-state the findings in a table.

We thank the reviewer for this suggestion and agree that the results section could be summarised by suppressing the overlapping data provided in the Tables. We have reformulated the results section accordingly.

Reviewer: 2

Thank you for allowing me to review this well-written manuscript. The authors attempt to create a clinical tool capable of determining who may be at risk of having poor outcomes after an initial ankle sprain. I think this study was well-done and has significant clinical implications. However, I do have some concerns and a few comments to assist in improving the manuscript for researchers and clinicians.

We thank the reviewer's kind words and hope to have addressed all of his concerns in the following responses and in the new version of the manuscript, as well as improved the reporting of our study based on his comments and suggestions.

Methods

Consider adding a flow diagram to both the patients included in the model as well as the process of selection of predictors. I think these visual representations may assist the reader in following the in-text iterative processes.

We thank the reviewer for this helpful suggestion. We have added a flow diagram detailing the predictor's selection process as a supplemental material. We did not include a flow diagram of the patients in the CAST trial though, as this has been previously published. Nevertheless, we have included a mention to it in the text.

156-157: I think a simple, independent samples t-test to look at differences in the data sets would be appropriate, especially due to the CAST including individuals "with an ankle sprain of grade 2 or 3" while the SPRAINED cohort included any ankle sprain.

We thank the reviewer for his suggestion. However, we have followed the TRIPOD Statement ^[1] (item 13c), which states 'For validation, show a comparison with the development data of the distribution of important variables (demographics, predictors, and outcome)' where the emphasis is to present the distribution of the predictors, demographics and outcome of both the development and validation data. Determining whether any predictors are statistically different at some arbitrary level of significance (and influenced by sample size) is limited to ascertain similarity or dissimilarity. From Table 1 it is quite evident which predictors are similar and which are not between the two data sets.

179-181: What was the rationale for selecting a p value of 0.157?

The decision to include predictors in the final model based on a p-value < 0.157 is that it is equivalent to Akaike Information Criteria, often shortened to AIC, a widely used criteria for variable selection was conservatively taken to prevent over fitting ^[2].

217-225: Going back to my previous point, I think this does warrant statistical tests to determine differences across groups.

We thank the reviewer for his comment. However, as stated in our response to an earlier comment, there is no rationale for calculating p-values to identify any statistically significant difference between any predictors in the two data sets (we refer the reader to the TRIPOD Statement, Explanation & Elaboration paper).

236-237 – I'm not sure if this is a valid enough of a reason to include BMI in the final model. While, BMI is a relatively simplistic way of measuring body composition, plenty of research has demonstrated its flaws. I'd like some more supportive objective commentary on its' inclusion, especially considering its relatively small contribution to the model.

We thank the reviewer for this comment and agree that further detailing on the rationale for the inclusion of BMI in the prognostic model would be important, as well as increase the completeness and transparency of our report. We have removed the brief comment on this matter from the results section and added the following paragraph to the manuscript's discussion.

“The inclusion of BMI in the prognostic model is another issue that deserves consideration. Although not statistically significant in the final multivariable logistic regression analysis, according to AIC ($p < 0.157$), we have decided to keep BMI in the model for several reasons. First, this decision prevented another round of predictor selection, which could increase over-fitting. The model building process was not solely based on statistical rationale, and BMI was considered to be an important predictor by clinicians during our consensus group meeting. BMI is an easy to assess surrogate measure of body weight that is frequently collected at clinical routine and one that most patients know how to calculate themselves. Finally, its inclusion does not add much complexity to the models.”

Moreover, after the reviewers' comments, we ran some additional analysis and found the following:

- BMI showed better predictive ability than weight itself in both unadjusted and adjusted logistic regression analyses.
- The model with BMI had better apparent performance, as estimated by the c-statistic, than the model without BMI or the model with weight replacing BMI.
- The model with BMI presented increased net benefit compared to the model without BMI or the model with weight replacing BMI.

Therefore, we hope to have allayed concerns this issued might have caused.

Discussion: I think the discussion could be developed a bit more. Some discussion on the individual predictors themselves would be appropriate.

We thank the reviewer for this helpful suggestion. Even though the aim of the study was not to identify and discuss individual prognostic factors of ankle sprain, but to develop and externally validate a model to predict poor recovery after this type of injury, we have added the following paragraph to the discussion.

“In a recent systematic review, we have reported that some of the variables selected for inclusion in our prognostic model, have been previously identified as important predictors of short, medium or long term recovery after ankle sprain.[8] According to O'Connor et al. age and weight bearing ability are predictors of ankle function, as measured by the Karlsson function score, both at 4 weeks and 4 months after injury.[38] Akacha et al. also demonstrated that age was an important predictor of slower and incomplete recovery after ankle sprain, as measure by the Foot and Ankle Outcome Score.[39] The magnitude of pain at rest at 3 months has also been shown to have prognostic value for poorer self-reported recovery at 12 months after ankle sprain by Van Middelkoop et al.[40] On the other hand, Findings regarding recurrence of ankle sprain are conflicting. McKeon et al., reported that recurrent ankle sprain was not a significant predictor of time to return-to-play after an ankle injury.[41]

This is contrary to reports of an association between recurrent sprains and chronic ankle instability reported in a systematic review conducted by Pourkazemi et al. [42] One possible explanation for these contradictory results may be the nature of the outcomes investigated in each study. When more subjective aspects of recovery (such as ankle function or instability) are considered in the definition of the endpoint, like in the present study, re-spraining the ankle seems to be an important predictor of recovery.”

We hope this will add some perspective to the discussion of our manuscript. However, we have opted to leave the focus of our discussion on the prognostic model as a whole and not to develop an in-depth discussion on the individual value of the selected predictors; which we have already addressed in the refereed systematic review published recently by our group.

VERSION 2 – REVIEW

REVIEWER	Adam Rosen USA, University of Nebraska at Omaha
REVIEW RETURNED	03-Jul-2018
GENERAL COMMENTS	The authors have done an exemplary job addressing my concerns, congratulations on an impactful manuscript. Please ensure consistency in reference formats (e.g. abbreviations/italicizing of journal names).